# Single-Molecule Mixture: A Concept in Polymer Science

**DOI:** 10.3390/ijms25147571

**Published:** 2024-07-10

**Authors:** Yu Tang

**Affiliations:** State Key Laboratory of Chemical Biology, Center for Excellence in Molecular Synthesis, Shanghai Institute of Organic Chemistry, University of Chinese Academy of Sciences, Chinese Academy of Sciences, Shanghai 200032, China; tangyu@sioc.ac.cn

**Keywords:** single-molecule mixture, polymer

## Abstract

In theory, two extreme forms of substances exist: the pure form and the single-molecule mixture form. The latter contains a mixture of molecules with molecularly different structures. Inspired by the “chemical space” concept, in this paper, I report a study of the single-molecule mixture state that combines model construction and mathematical analysis, obtaining some interesting results. These results provide theoretical evidence that the single-molecule mixture state may indeed exist in realistic synthetic or natural polymer systems.

## 1. Introduction

Realistic substances always contain a mixture of molecules, and, in theory, two extreme forms of substances—a pure form and a single-molecule mixture form—exist, as shown in Figure 1. The former contains only one kind of molecule, while the latter comprises a mixture of molecules with molecularly different structures. To the best of my knowledge, this elusive molecular form has not yet been explored in the literature. When considering a “single-molecule mixture”, two interesting questions arise: (1) Do “single-molecule mixture”-state molecules exist in nature or synthetic systems? (2) Is there any possible way to synthesize “single-molecule mixture”-state molecules? To explore these questions, I have herein attempted to establish a theoretical approach, combining model construction and mathematical analysis, to study this molecular state and provide theoretical evidence supporting the conclusion that these mixtures may indeed be widely distributed in nature or in synthetic polymer systems. In addition, a possible synthetic route to “single-molecule mixture”-state molecules has also been provided.

Since the beginning of the 21st century, the concept of “chemical space” has gradually attracted the attention of the scientific community [1,2,3,4]. Chemical space—encompassing all possible small organic molecules, including those present in biological systems—is extremely vast. For example, Dobson^1^ estimated that, seeing as there are 20 different amino acid types and the average natural protein comprises about 300 residues, the overall number of possible isomers is a staggering 20^300^, or even more than 10^390^, and, if a single molecule of each of these polypeptides were to be produced, the combined mass would vastly exceed that of the known universe. Inspired by these remarkable results, I envisioned that, if I first constructed a model polymer space containing an extreme number of possible isomers and then (randomly) synthesized molecules with only a tiny fraction of this space’s structures, I would have a very high probability of obtaining a “single-molecule mixture”-state molecule. This consideration constitutes the basic design principle of the current study.

## 2. Results and Discussion

To start this exploration, a model polymer molecule system is first constructed, as illustrated in Figure 2, where the overall number of substitutable sites is *a*, and each site is randomly substituted by one of the two possible groups, R^1^ or R^2^. Thus, the overall number of structural isomers in this polymer space is 2*^a^*, exponentially increasing as *a* increases. For example, if *a* = 200, then the overall number of isomers in this space is 2^200^—namely, 1.60693804 × 10^60^.

A very interesting question arises: if I were to take a given number (*n*) of structures from the above structural space (overall number of structures = *m*) individually and randomly (meaning that the same structure may be taken more than once), how would I calculate the probability (*P*) of obtaining a single-molecule mixture (where each structure is different)?

This problem can be solved by calculating probability.
(1)$$P=\frac{C_{m-n}^{n}}{C_{m}^{n}}=\frac{\left(m-n\right)\left(m-n-1\right)\left(m-n-2\right)\cdots \cdots (m-n-n)}{m\left(m-1\right)\left(m-2\right)\cdots \cdots (m-n)}\;=(1-\frac{n}{m})(1-\frac{n}{m-1})\;(1-\frac{n}{m-2})\cdots \cdots (1-\frac{n}{m-n})\;\mathrm{If}\;m\;>\;2n,\;\mathrm{then},\;P\;>\;(1+\frac{n}{n-m}{)}^{n}\;\mathrm{and}\;\frac{n}{n-m}\;-1$$

Then, one can use Bernoulli’s inequality to approximate (1 + $\frac{n}{n-m}$)^n^ as follows:$$(1+\frac{n}{n-m}{)}^{n}\;>\;1+n\times \frac{n}{n-m}$$

In this case, I chose *n* = *N*_A_ (6.02 × 10^23^) as the given number of molecules taken from the structural space and *a* = 200 as the number of substitutional sites in the polymer model system (thus, *m* = 2^200^ = 1.60693804 × 10^60^). As a result, the probability (*P*) of obtaining a single-molecule mixture is the following:$$P\;>\;(1+\frac{n}{n-m}{)}^{n}\;>\;1+n\times \frac{n}{n-m}\;\approx \;1\;-\;2.26\;\times \;10^{-13}$$

This result reveals that, if we were to individually and randomly take *n* = *N*_A_ (6.02 × 10^23^) molecules from this model structure space (*a* = 200, *m* = 1.60693804 × 10^60^), the probabilities (*P*) of obtaining single-molecule mixtures would be close to 1.

According to Equation (1), the relationship between *P* and *m* is
(2)$$m<\;n+n^{2}\times \frac{1}{1-P}$$

From Equation (2), we can estimate that for a given *P*, the minimum number of structural isomers *m* in the model structure space should exceed n + n^2^ $\times \frac{1}{1-P}$.

It has been noted that, in many realistic synthetic polymer systems and natural biopolymer systems, the partial random substitution of the parent molecule by a few substituent groups frequently occurs, such as in the polymer bromination process [5,6] (random partial substitution of the hydrogen atom by a bromine atom), many other post-polymerization modification processes [7,8,9,10,11,12,13,14,15], and the DNA [16] and protein methylation processes (random partial substitution of the hydrogen atom by a methyl group) [17,18], as illustrated in Figure 3 and Appendix A. This substitution process generates a large number of isomers with equal probabilities. In order to calculate the exact number of potential isomers, I constructed two additional model systems, as illustrated in Figure 4.

If the overall number of substitutable sites in the parent molecule is *m*, that of the substituted sites is *n*, and that of the possible substitute groups in each randomly substituted site is *s*, then the overall number of potential isomers from this structural space *r* can be calculated as follows:(3)$${r=C}_{m}^{n}\times \;s^{n}$$

In model system I, I chose *m* = 1000 and *s* = 1, implying that the parent molecule contained 1000 substitutable sites. If a few of these sites are randomly substituted by another group (R^0^ → R^1^), the overall number of potential isomers can be calculated as follows:(4)$${r=C}_{1000}^{n}$$

In model system II, I chose *m* = 100 and *s* = 10, implying that the parent molecule contains 100 substitutable sites. If a few of these sites are randomly substituted by one of the ten possible R groups [R^0^ → (R^1 to 10^)], the overall number of potential isomers can be calculated as follows:(5)$${r=C}_{100}^{n}\;\times \;10^{n}$$

The calculated overall numbers of potential isomers (*r*) (*n* = 0–25) are listed in Appendix A, and the relationship between lg*r* and *n* is depicted in Figure 5.

From Figure 5, we can see that, for these two model systems, the overall number of potential isomers exponentially increases alongside the number of substituted sites. If *n* = 25, then the overall number of potential isomers is 4.76 × 10^49^ and 2.43 × 10^48^ for model systems I and II, respectively. According to Equation (1), when taking 1 mmol (6.02 × 10^20^) molecules from these two structural spaces, the probability (*P*_1_ and *P*_2_ for model systems I and II, respectively) of obtaining a single-molecule mixture is the following:$$P_{1}\;>\;(1+\frac{n}{n-m}{)}^{n}\;>\;1+n\times \frac{n}{n-m}\;\approx \;1-7.61\;\times \;10^{-9}\;\mathrm{and}\;P_{2}\;>\;(1+\frac{n}{n-m}{)}^{n}\;>\;1+n\times \frac{n}{n-m}\;\approx \;1-1.43\;\times \;10^{-7}$$

These calculated results indicate that these two probability values are sufficiently large for single-molecule mixtures to exist at millimole-scale levels.

According to Equation (2), we can estimate that if we take 1 mmol (6.02 × 10^20^) molecules from a structural space, the probability of obtaining a single-molecule mixture state exceeds 0.999; thus, the minimum number of structural isomers *r* in the model structure space should exceed n + n^2^ $\times \frac{1}{1-P}$ ≈ 3.62404 × 10^44^; this value is significantly smaller than the actual isomer numbers of the above two spaces.

Thus, we can conclude that, if a polymer molecule contains 1000 substitutable sites and 2.5% of these sites are randomly substituted by another group, the potential isomers exceed 4.76 × 10^49^, and, if this substitution process proceeds at a millimole scale, then the products most likely exist in a single-molecule mixture state.

Finally, to better illustrate the structural features of single-molecule mixture-state polymers, a 24 mer *O*-propyl-substituted D-mannitol model system was built, as illustrated in Figure 6A (A possible synthetic route toward this 24 mer was provided in Appendix A. see Appendix A). This system contained 192 phenolic hydroxyl groups, randomly substituted by the *n*-propyl or *i*-propyl group. This system’s molecular weight was 23,775.1700 Da, and the number of structural isomers was 2^192^—namely, 6.277 × 10^57^. According to Equation (1), even if we were to prepare this polymer at a 100 mol scale (ton-scales), the probability (*P*) of obtaining a single-molecule mixture would be the following:$$P\;>\;(1+\frac{n}{n-m}{)}^{n}\;>\;1+n\times \frac{n}{n-m}\;\approx \;1\;-\;5.77\;\times \;10^{-7}$$

In this model system, even though each molecular structure was different, as the R group in each substitute site varied, they all came randomly from the same structural space, inspired by the concept of orbital hybridization [19,20,21,22]. If we were to define Pr* as a hybrid structure of *n*-Pr and *i*-Pr, then this polymer system could be viewed as a “pure substance system”, in that each phenolic hydroxyl group would be substituted by an unprecedented, hybridized Pr* group, as illustrated in Figure 6B. Thus, certain physical proprieties of this polymer system would be expected, to some extent, to be similar to those of a pure substance system. However, the impact of the single-molecule state on the physical, chemical, and biological properties of natural and synthetic polymers remains a subject of further research.

## 3. Materials and Methods

The work depicted in this paper is a theoretical research that combines model construction, mathematical analysis, and thought experiments. Model construction was performed using ChemOffice software (PerkinElmer ChemOffice 2020–Version 20.0), probability approximation was performed using Bernoulli’s inequality, detailed calculation was performed on a household computer using free online scientific calculator (Desmos, https://www.desmos.com/scientific?lang=zh-CN, accessed on 8 July 2024).

## 4. Conclusions

In summary, single-molecule mixtures, opposite to the “absolute pure” extreme molecular state, were theoretically studied by combining model construction and mathematical analysis, obtaining some interesting results. These results support the conclusion that the single-molecule mixture state may be widely distributed in realistic synthetic or natural polymer systems.

Chemical microheterogeneity—the quality of being heterogenous at the microscopic level—is an attribute of all living systems and most soft and crystalline materials [23,24]. According to a recent classification by Gentili et al. [23], natural and synthetic polymer systems are one of the five major sets of microheterogeneous systems. While it has long been recognized that microheterogeneity exerts a major influence on the function of biopolymers such as glycoproteins [25], the concept of a “single-molecule mixture” and the results of the current study shall provide a unique way of understanding “microheterogeneity” at the molecular level. It is hoped that this study will inspire further exploration of this intriguing area.

## Figures and Tables

**Figure 1 ijms-25-07571-f001:**
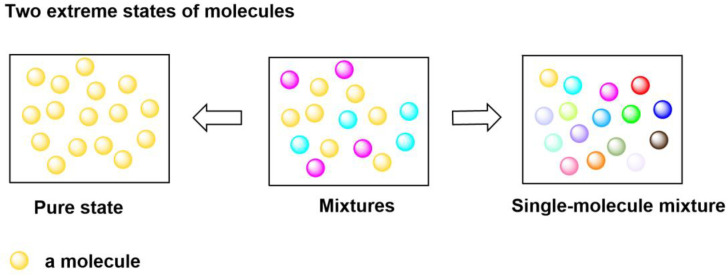
Two extreme states of substances: the pure state and the single-molecule mixture state. Different colors of balls represent molecules with different structures.

**Figure 2 ijms-25-07571-f002:**
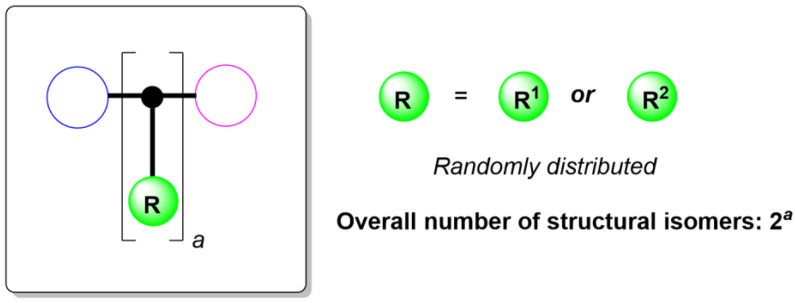
A model polymer molecule system.

**Figure 3 ijms-25-07571-f003:**
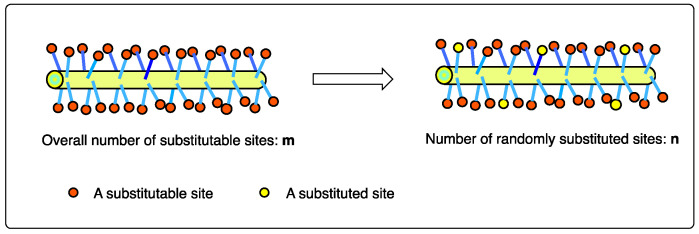
The partial random substitution process of a parent molecule containing m substitutable sites by n substituent sites.

**Figure 4 ijms-25-07571-f004:**
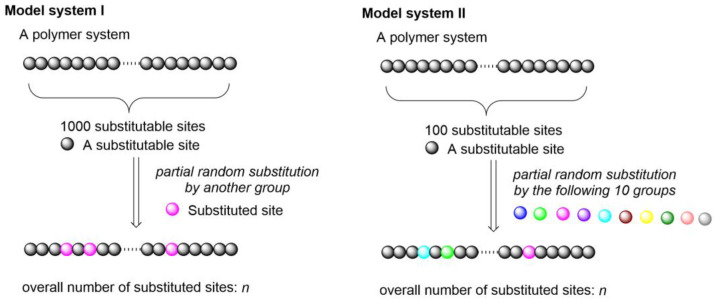
Schematic diagram showing model systems I and II. Different colors of balls represent different substitution groups.

**Figure 5 ijms-25-07571-f005:**
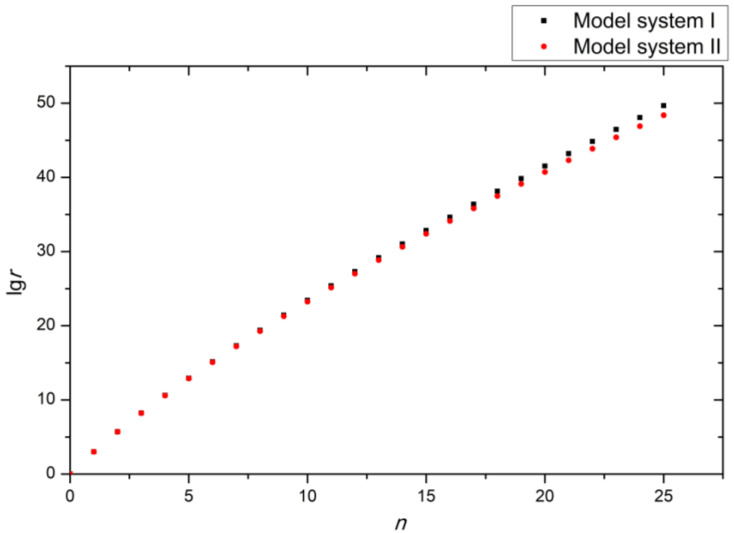
The relationship between lg*r* and *n.*

**Figure 6 ijms-25-07571-f006:**
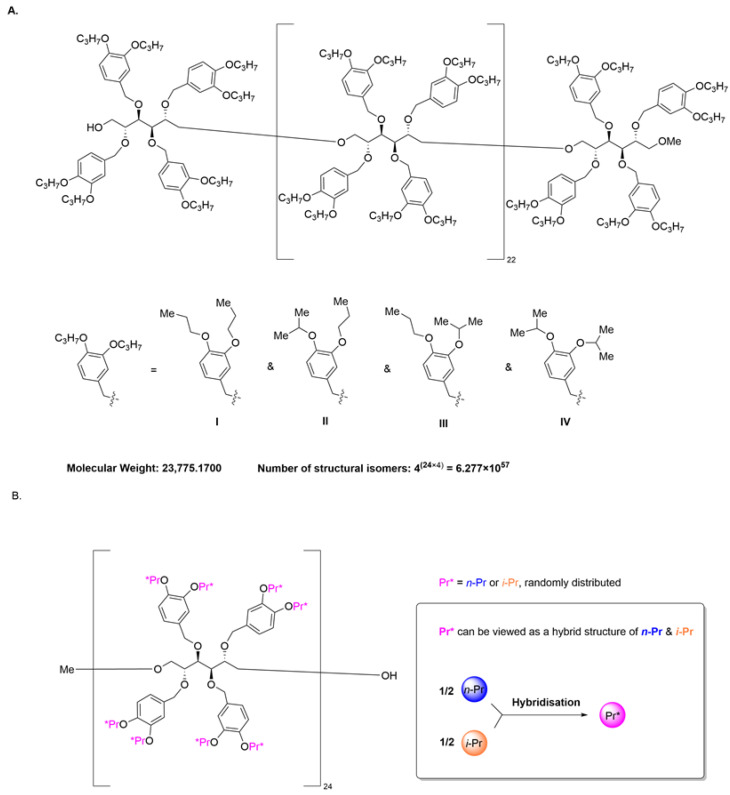
(**A**) A 24 mer *O*-propyl-substituted D-mannitol model system. (**B**) Hybrid molecular structure of the single-molecule mixture model system.

## Data Availability

The original contributions presented in the study are included in the article/Appendix A, further inquiries can be directed to the corresponding author.

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
