# Peer review of "Single-Molecule Mixture: A Concept in Polymer Science"

_ijms, 2024, doi:10.3390/ijms25147571_

Round 1

Reviewer 1 Report

Comments and Suggestions for Authors

The manuscript from Tang, Y., introduces a concept of single molecule mixture in a polymer system, where each molecule has a unique structure. The author establishes a mathematic model to predict the possibility of achieving such a state. While the approach to understanding the complexity of polymer materials is interesting, the manuscript lacks a comprehensive background introduction, the methodology is not well-explained, and the specific applications or insights of the finding are not discussed. 

Below are some points for the authors to consider.

(1)  In the abstract and introduction sections, the author does not introduce other works in the same field, limiting the readers’ understanding of the study’s impact. An overview of related literatures would help on highlighting its significance. 

(2)  The manuscript uses “we” despite having only one author. It should be changed to “I” to reflect the single authorship. 

(3)  In lines 40-45, the probability calculation is difficult to follow. The author should provide more detailed explanations to enhance clarity.

(4)  Figure 3 caption is confusing. In Figure 3, n represents “number of randomly substituted sites” while in the caption, n represents the number of “substituent groups”. What is the difference between “site” and “group”?

(5)  To make the data more accessible, I suggest showing two plots for the model systems I and II instead of using Tables 1 and 2.

(6)  In the conclusion section, the discussion about the study’s impact is superficial. The author should elaborate on the property differences among “pure form”, “mixtures” and “single-molecule mixture” and explain the insights that the finding on single-molecule mixtures provide.

Author Response

Please see the atttachment.

Reviewer 2 Report

Comments and Suggestions for Authors

Comments on the Quality of English Language

Minor revisions on syntax and grammar throughout the paper must be done and in some parts, some statements can be better expressed with a better choice of words. 

Round 2

Reviewer 1 Report

Comments and Suggestions for Authors

The author has successfully addressed my previous concerns. The revised introduction and conclusion are clear, significantly improving the quality of the manuscript. 

Reviewer 2 Report

Comments and Suggestions for Authors

The author made extensive revisions which addressed the majority of the concerns raised in the first round of review, especially with the addition of figures which supported the discussion furhter. However, I still have some few questions that may be regarded as minor revisions.

1. With regards to my Question #3 in the first round, it is good that P1 and P2 values which correspond to probabilities of getting a single-molecule mixture in models 1 and 2 are presented.

However, in the latest revision there is this statement "These calculated results indicated that these two r values are..." which was stated after presenting the probabilities - are the r values being referred to here P1 and P2? What do these r values represent? If they refer to the probabilities, then it is better to refer to them simply as probabilities instead of r values.

2. Also as a follow up to the question above, I still do not understand the criteria for the claiming "then the potential isomers exceed 4.76×1049, if this substitution process proceed at millimole-scale, then the products are most likely existing as single-molecule mixture state!”.

What is the basis for saying that this particular number (4.76×1049) indicates that products can exist in single-molecule scale? If you can provide any reference or standard (for example, a reference saying that a number >1049 is a criterion), then I'd think it would help in making the readers understand the points you would like to convey.

Comments on the Quality of English Language

There are still a few parts of the manuscript that needs minor English editing (better choice of words and some grammatical inconsistencies).
